# A Multicenter Study to Assess a Systematic Screening of Occupational Exposures in Lung Cancer Patients

**DOI:** 10.3390/ijerph20065068

**Published:** 2023-03-13

**Authors:** Olivia Pérol, Nadège Lepage, Hugo Noelle, Pierre Lebailly, Benoit de Labrusse, Bénédicte Clin, Mathilde Boulanger, Delphine Praud, Françoise Fournié, Géraud Galvaing, Frédéric Dutheil, Brigitte Le Meur, Daniel Serin, Eric Dansin, Catherine Nisse, Barbara Charbotel, Beatrice Fervers

**Affiliations:** 1Département Prévention Cancer Environnement, Centre Léon Bérard, 69373 Lyon, France; 2INSERM U1296 Radiations: Défense, Santé, Environnement, Centre Léon Bérard, 69373 Lyon, France; 3CHU Lille, Centre Régional de Pathologies Professionnelles et Environnementales, 59800 Lille, France; 4CHU Lille, ULR 4483-IMPECS-IMPact de l’Environnement Chimique sur la Santé Humaine, University of Lille, 59000 Lille, France; 5Faculté de Médecine, Université Claude Bernard Lyon 1, 69373 Lyon, France; 6Service d’Evaluation Economique en Santé, Hospices Civils de Lyon, Pôle de Santé Publique, 69003 Lyon, France; 7INSERM U1086, Unité de Recherche Interdisciplinaire pour la Prévention et le Traitement des Cancers, Université de Caen Normandie, UFR Santé, 14032 Caen, France; 8Centre François Baclesse, 14076 Caen, France; 9Institut du Cancer Sainte Catherine, 84918 Avignon, France; 10Service de Santé au Travail et Pathologie Professionnelle, CHRU de Caen, 14033 Caen, France; 11Département Interdisciplinaire des Soins de Support du Patient en Oncologie, Service Social, Centre Léon Bérard, 69373 Lyon, France; 12Chirurgie Thoracique, Centre Jean Perrin, 63011 Clermont-Ferrand, France; 13Université Clermont Auvergne, CNRS, LaPSCo, Service Santé Travail Environnement, CHU Clermont Ferrand, 63011 Clermont-Ferrand, France; 14Département d’Oncologie Médicale, Centre Oscar Lambret, 59000 Lille, France; 15UMRESTTE (Unité Mixte IFSTTAR/UCBL), Université Lyon 1, 69373 Lyon, France; 16Service des Maladies Professionnelles, Hospices Civils de Lyon, Centre Hospitalier Lyon Sud, 69495 Pierre-Bénite, France

**Keywords:** occupational exposures, lung cancer, systematic screening, compensation

## Abstract

Occupational lung cancer cases remain largely under-reported and under-compensated worldwide. In order to improve the detection and compensation of work-related lung cancers, we implemented a systematic screening of occupational exposures, combining a validated self-administered questionnaire to assess occupational exposures and a specialized occupational cancer consultation. After a pilot study, the present prospective, open-label, scale-up study aimed to assess this systematic screening of occupational exposures in lung cancer patients in five sites in France by associating university hospitals with cancer centers. Patients with lung cancer were sent a self-administered questionnaire to collect their job history and potential exposure to lung carcinogens. The questionnaire was assessed by a physician to determine if a specialized occupational cancer consultation was required. During the consultation, a physician assessed if the lung cancer was occupation-related and, if it was, delivered a medical certificate to claim for compensation. Patients were offered help from a social worker for the administrative procedure. Over 15 months, 1251 patients received the questionnaire and 462 returned it (37%). Among them, 176 patients (38.1%) were convened to the occupational cancer consultation and 150 patients attended the consultation. An exposure to occupational lung carcinogen was identified in 133 patients and a claim for compensation was judged possible for 90 patients. A medical certificate was delivered to 88 patients and 38 patients received compensation. Our national study demonstrated that a systematic screening of occupational exposures is feasible and will bring a significant contribution to improve the detection of occupational exposures in lung cancer patients.

## 1. Introduction

Lung cancer is the second most common cancer after breast cancer, with 2,206,771 new cases (11.4% of all new cancers), and the leading cause of cancer death, accounting for nearly one in five cancer deaths (1,796,144 deaths reported in 2020 worldwide) [1]. Tobacco exposure is a well-known carcinogen, and more than 80% of lung cancers are estimated to be attributable to tobacco smoking [2]. However, numerous other well-established carcinogenic agents are causally associated with lung cancer etiology [3]. Overall, in 2017, the International Agency for Research on Cancer (IARC) identified 19 IARC Group 1 occupational lung cancer carcinogens (substances or mixtures) as well as eight occupations, industries, or work processes [3]. Lung cancer is by far the most common cancer associated with occupational carcinogenic exposures [4]. In France, almost 15% (i.e., close to 6000 cancer cases; 19.6% for men and 2.6% for women) have been estimated to be attributable to occupational exposures [5,6]. Similar population attributable fractions, estimated from the prevalence of occupational exposures to recognized lung carcinogens and corresponding risk estimates, have been reported for other countries, in particular Great Britain and Canada, where equally 15% of all lung cancers have an occupational contribution [7,8]. Moreover, in occupational populations, multiple lung cancer risk factors frequently coincide, and synergistic effects have been reported between tobacco and some occupational exposures [9,10,11].

Despite European regulations on occupational exposure limit values and workers protection for carcinogenic agents, there is a large diversity of national compensation systems and practices to compensate occupational cancers [12]. While asbestos-related lung cancers are predominant among compensated cancers [12], due to national reporting laws and social solidarity schemes in many industrialized countries [13], occupational lung cancer cases remain largely under-reported and under-compensated worldwide [14,15,16,17,18,19].

In France, while the number of recognized occupational cancers has tripled in 20 years and the ratio of recognized occupational cancers (i.e., 11.39) is the second highest in Europe after Germany [12], there are less than 2000 cancers recognized annually, 57% being lung cancers [20]. It has been estimated that 60% of occupational lung cancers are not compensated [21].

Under-reporting of occupational cancers can be explained by the lack of awareness and expertise of physicians to assess occupational exposures [16,22,23], as well as the time and effort required to collect the occupational history and to report, which are often difficult to reconcile with the clinical workload [14,23]. Moreover, the high prevalence of smoking in lung cancer patients has been identified as a barrier to the identification and reporting of occupational lung cancers [24,25]. Several studies assessing interventions in physician practices have reported that they did not substantially improve reporting of occupational diseases [14,16,24,26]. Furthermore, the long latency between exposures and lung cancer development, as well as the absence of impact of occupational exposures on cancer treatment, have been stressed as reasons for under-recognition and under-compensation of work-related lung cancers.

Under-compensation can be further explained by the complexity of administrative procedures [27,28,29], which is emphasized in the context of concomitant cancer treatments, fatigue, poor prognosis, and a limited knowledge of past occupational exposures in patients (often with a low education level) [27,30]. Studies have shown that some factors, such as being a man aged <65 years at diagnosis, have been found to be significant predictors of a higher probability to receive compensation [31].

To overcome these barriers and in order to improve the detection and compensation of work-related lung cancers, we implemented a systematic screening of occupational exposures, combining a validated self-administered questionnaire (SAQ) to assess occupational exposures and a specialized occupational cancer consultation [29,32]. We successfully completed a pilot study among 440 lung cancer patients in a comprehensive cancer center in France, demonstrating the feasibility and the capacity of the systematic screening to improve detection and compensation in this population with a limited cost (EUR 62 per lung cancer patient) [29]. The proportion of occupational lung cancers identified was consistent with that expected from the literature [6,29].

The present scale-up study aimed to assess the implementation of this systematic screening of occupational exposures at a national level in order to confirm its feasibility and capacity to improve the detection of occupational exposures and the compensation of occupational lung cancers.

## 2. Materials and Methods

The study protocol was approved by the French ethics committee “CPP Ile de France X” (ID RCB: 2017-A00349-44) and the study database was reported to the National Commission for Data Protection and Liberties (CNIL) (reference number: 2016181 v0).

### 2.1. Study Design

PROPOUMON national was a prospective, national, multicenter, open-label study promoted by the Léon Bérard regional comprehensive cancer center (Lyon, France).

Patients were recruited in five investigating cancer centers (Lyon, Caen, Lille, Clermont-Ferrand, and Avignon). This study was conducted in close collaboration with the occupational diseases consultation center in university hospitals in four sites (Lyon, Caen, Lille, and Clermont-Ferrand). In three sites (Caen, Lille, and Clermont-Ferrand), the occupational cancer consultation took place in the university hospital, and in the two remaining sites, the occupational cancer consultation took place in the cancer center. The study teams in each site further involved a clinical research assistant (in the cancer center) and a social worker (in the cancer center or the university hospital, depending on the local organization).

### 2.2. Study Population

Patients aged ≥18 years old with a histologically confirmed lung cancer and treated in one of the five investigating centers were eligible. Patients managed elsewhere, or managed in the investigating centers only for radiotherapy, diagnostic procedures, or medical second opinion, were not eligible. Patients who had already attended an occupational cancer consultation as well as patients who were not able to read, write, and understand French were also not eligible for this study.

### 2.3. Systematic Screening Procedure

The systematic screening procedure was identical to the pilot study [29]. Briefly, eligible patients were identified through the weekly multidisciplinary lung cancer board in each participating cancer center. They were sent an information letter, a self-administered questionnaire for occupational exposures screening (SAQ), and the Evaluation of Deprivation and Inequalities in Health Examination Centers (EPICES) questionnaire with a prepaid return envelope. The purpose of the SAQ was to collect information about the patients’ education level, job history (job title, start and end dates, employer, sector of activity, and tasks performed for each occupation) as well as self-reported occupational exposures from a list of 25 lung carcinogens and corresponding occupation. The EPICES questionnaire (assessing marital status, health insurance status, economic status, family support, and leisure activity) was used to assess individual deprivation [33]. Patients not having returned the questionnaires were contacted by the clinical research assistant, who offered help to complete them. Considering the average delay for returning the SAQ experienced in the pilot study (i.e., 47 days), the time interval from sending the SAQ to the reminder call was extended to one month for the present study (compared to three weeks in the pilot study). At reception, the SAQ was assessed by a physician to determine if a specialized occupational cancer consultation was required, based on the jobs, tasks, and exposures reported by the patient. The evaluations were performed independently in each recruiting center by a physician with expertise in occupational health. When no occupational exposure was identified, the patients received a personalized letter informing them that their disease was assessed as not work-related.

### 2.4. Occupational Cancer Consultation

During the consultation, the physician collected data on the patient’s job history, exposure to carcinogens, conditions, frequency, duration and level of exposure, means of protection, and non-work-related risk factors (i.e., smoking history and non-occupational exposure, in particular to asbestos). If the physician considered the lung cancer could be occupation-related, a medical certificate, required for compensation claims, was delivered. Patients exposed to asbestos were given a medical certificate to make a claim to the French asbestos victim compensation fund (FIVA). Patients who wanted to claim for compensation were offered help from a social worker for the administrative procedure. Unlike the pilot study, the social worker carried out a 6-month follow-up by phone to assess with the patient the state of progress of the compensation procedure and to provide further assistance if necessary.

### 2.5. National Multidisciplinary Occupational Cancer Board

Given the complex nature of occupational exposures, particularly in patients with numerous consecutive employments who were exposed to multiple carcinogens and the complexity of the administrative procedures, an occupational cancer board was set up to gather all participating centers for monthly meetings, with two main objectives: to exchange on professional practices, and to discuss in depth complex situations to facilitate compensation claims.

### 2.6. Data Collection

In addition to data collected through the questionnaires (SAQ and EPICES), the physician’s assessment on the link between occupational exposure and the lung cancer was collected from the occupational cancer consultation, including the degree of imputability (i.e., the level of certainty that the lung cancer is related to occupational exposure) based on the occupations held and job activities, employment dates, type of exposures, and their intensity, frequency, and duration [34].

The imputability was classified by the physician according to three levels:–“low” (exposure not substantial or difficult to quantify; not eligible for compensation);–“moderate” (significant exposure that did not meet all the criteria of occupational lung cancer but eligible for compensation claim; presence of extra-professional risk factor (i.e., tobacco smoking));–“high” (substantial exposure meeting the criteria for occupational-exposure-related lung cancer).

Demographic characteristics including birth date and gender, extra-professional lung cancer risk factors including smoking history, lung cancer history (date of diagnosis, patients’ status: newly diagnosed/under follow-up/progressive disease), tumor characteristics (histology, cancer stage), and cancer treatments were also collected. All clinical data were extracted from the patients’ electronic medical records.

Process data on the screening intervention itself were collected at every step (SAQ, reminder call, occupational cancer consultation, and compensation process).

### 2.7. Statistical Analysis

All enrolled patients were included in the analysis.

Participants’ characteristics were described using means, standard deviations (SD), and minimum and maximum for quantitative data and were described with frequencies and percentages for qualitative data. Demographic characteristics and clinical data were compared between centers using t-tests or Wilcoxon rank sum test for quantitative data and chi-squared or Fisher’s exact test for qualitative data. Patient characteristics related to participation of the screening process were also compared. For all statistical tests, *p* values < 0.05 were considered statistically significant. Statistical analyses were performed using R software (version 4.0.2).

## 3. Results

Between December 2017 and March 2019, 2900 patients were screened in the five participating comprehensive cancer centers. Overall, 1251 patients (43.1%) met the eligibility criteria and were included in this study. The proportion of non-eligible patients varied by center (42.5% in Caen, 23.8% in Lyon, 19.2% in Clermont-Ferrand, 14.3% in Lille, and 0.1% in Avignon). Reasons for non-eligibility are presented in the study flowchart (Figure 1).

### 3.1. Patients’ Characteristics

Patients’ characteristics are summarized in Table 1 (demographics) and Table 2 (clinical data). Patients in the study population were predominantly men (65%) with a mean age of 66 years (SD = 10.1) with no statistical differences between centers in demographic data. Low education level was frequent, with 26% of patients having no diploma, although disparities existed between centers. Concerning vulnerability, 36% of patients were identified as socially deprived, with a mean EPICES score of 26 (SD = 18.9). The proportion of socially deprived patients differed statistically between centers (*p* = 0.015), with the lowest proportion in Clermont-Ferrand (23.3%) and the highest in Caen (54%). Most patients (85.1%) were active smokers (current and former smokers) with a mean tobacco consumption of 40.6 pack-years (SD = 20.8). The proportion of non-smokers differed between centers (*p* < 0.001) and ranged from 8.7% in Avignon to 19% in Lyon (Table 1).

The majority of patients were enrolled at lung cancer diagnosis (57%), especially in centers where an occupational consultation was in place prior to this study (Lyon, Caen, and Avignon). Overall, adenocarcinoma was the most prevalent histology (62.2%), and more than half of patients had metastatic lung cancer at diagnosis (56.1%) with a significant difference between centers (*p* < 0.001), ranging from 44% stage IV patients in Clermont-Ferrand to more than 60% in Avignon.

### 3.2. Systematic Screening of Occupational Exposures Process

Figure 1 details the screening process of occupational exposures with reasons for drop-out at each step. The SAQ was sent to 1249 patients (99.8% of study participants) and returned by 462 patients (37%). The questionnaire was not sent to two eligible patients who experienced a major clinical deterioration between the screening at the multidisciplinary lung board and the sending of the questionnaire. The response rate varied significantly (*p* < 0.001) between centers, from 23.1% in Avignon to 47.9% in Lille (Table 3). Patients aged ≥65 years and men returned the SAQ more than younger patients and women (Table 4). Smokers responded less to the SAQ than former smoking patients and non-smokers (*p* < 0.001). Patients with a localized disease returned the SAQ more frequently than metastatic patients (*p* = 0.028). The time of enrollment did not impact the SAQ return.

The average delay for returning the SAQ was 40.9 days (SD = 38.8). Overall, 953 reminder calls were performed; 193 patients (24.5%) could not be contacted by phone despite three attempts. Among the 787 patients who did not complete the SAQ, 185 patients (23.4%) reported to feel unconcerned by occupational exposures, 76 were deceased at the time of the reminder call (9.6%), and 41 patients (5.2%) refused to fill in the SAQ; for the remaining 485 patients (61.6%), non-response reasons were not further specified.

Based on the SAQ assessment by a physician, 176 patients (38.1%) were convened to the occupational cancer consultation. The proportion of patients for whom an occupational cancer consultation was considered relevant differed substantially between sites (*p* = 0.001), ranging from 19% of patients having returned the SAQ in Clermont-Ferrand to 55% in Avignon (Table 3).

### 3.3. Occupational Cancer Consultation

Among the 176 patients invited for an occupational cancer consultation, 150 (86.2%) attended the consultation. The reasons for non-attendance were patient’s death (*n* = 7), patient’s refusal (*n* = 5), and other reasons not further specified (*n* = 14). The mean delay between the SAQ return and the occupational cancer consultation was 73 days (SD = 52.3) with important variations between sites, ranging from 48 days (SD = 37) in Avignon to 107 days (SD = 36) in Caen.

During the consultation, an exposure to at least one occupational nuisance was identified in 133 patients (89%), including 130 patients (87%) exposed to occupational lung carcinogens (only the main occupational exposure was considered here for compensation purposes). The most prevalent main exposure was asbestos (*n* = 95); other exposures involved diesel fumes (*n* = 6), ionizing radiations (*n* = 5), coal and derivatives/polycyclic aromatic hydrocarbons (*n* = 5), welding fumes (*n* = 4), polymers (*n* = 4), second-hand smoke (*n* = 3), silica (*n* = 2), chromium (*n* = 2), and more marginally (*n* = 1), exposure to wood dust, arsenic, chromates, ether, radon, pesticides, and sulfuric acid. The level of imputability was considered high for 41 patients (30.8%), moderate for 59 patients (44.3%), and low for 26 patients (19.5%). The occupational exposures of the seven remaining patients (5.2%) were considered to be not related to lung cancer.

A claim for compensation was estimated possible under the French system for 90 patients (60%). A medical certificate, required for compensation claim, was delivered to 88 patients; two patients declined the deliverance of a medical certificate. The proportion of patients to whom the physician delivered a medical certificate at the end of the consultation varied considerably among study sites, ranging from 41% in Lyon to 100% in Caen and Clermont-Ferrand (Table 3). The main occupational exposure considered for compensation claim was asbestos (89.8%) (Table 5). The disease was considered as non-work-related for 33 patients (22%) and a compensation claim was judged impossible or unlikely to be successful under the French system for 12 patients (8%) (Figure 1).

Regarding compensation, 65 patients (73.9%) submitted a claim: 38 patients obtained compensation (36 related to asbestos exposure), 13 claims were rejected, and 14 were still under assessment by the relevant national bodies at the time of the analysis (Figure 1).

Concerning the national multidisciplinary occupational cancer board, 13 meetings were conducted throughout the study period, with the regular involvement of four centers and a total of 53 cases discussed.

## 4. Discussion

The objective of this study was to confirm our pilot study results in structures with a diversity of patients and different contexts of medical care to ensure its efficacy and reproducibility. The whole process was well-deployed in the participating centers (inclusions, management of SAQ, consultation, compensation process). The results confirmed the differences in the case mix and the disparities among patients regarding demographic and socio-economic characteristics. As our results show that participation in the screening process was statistically associated with the age and the sex of the patient, the lung cancer stage, and the smoking status, these disparities may have an impact on the level of patient information regarding occupational exposures and, thus, on patient acceptability to the screening process.

Our study confirmed the frequency of occupational exposures in lung cancer patients, as 19% of patients who returned the SAQ were eligible to claim for compensation under the French system (i.e., 7% of all enrolled patients). Our findings are similar to the results of the pilot study to assess the feasibility of a systematic screening for occupational exposures [29] and the current literature concerning estimates of the fraction of lung cancers attributable to occupational exposures [5]. Our systematic screening of occupational exposures in lung cancer patients improves the identification of occupational lung cancers. Yet, we could have expected a higher rate of compensation for occupational lung cancers. Hence, only 3% of our overall population was compensated (i.e., 8% of patients who responded to the SAQ). While the ratio of recognized cases compared to the claims for recognition in France was 79% in 2016 [12], the ratio was 58.4% in the present study. In our results, we have focused only on the main occupational exposure. In the case of exposure to asbestos and other carcinogens simultaneously, the exposure to asbestos was systematically prioritized, as in France, compensation is more easily obtained for asbestos exposure due to the existence of the asbestos victim compensation fund.

Despite the wide acknowledgement of under-reporting and under-compensation of occupational lung cancers, few actions to improve this issue in the clinical setting have been conducted. An Italian team carried out a process involving physicians to refer incident lung cancer to an Occupational Health Unit, which then conducted an interview to assess occupational exposure [26]; a similar experiment was conducted in Spain for several occupational diseases, which helped to improve reporting and official recognition [35]. Morell et al. offered a systematic occupational consultation to all patients with lung cancer conducted by a trained physician [16], while in Norway, a team collected occupational histories from a register [36]. All these initiatives helped to improve the reporting of occupational lung cancers. Nevertheless, in order to replicate and generalize a method, it is important to consider a straightforward, reproducible process that is feasible given the often heavy clinical workload.

While the population of the present study was composed of 35% women, the compensation medical certificates were almost exclusively delivered to men (97%). Given the large difference between men and women regarding lung cancers attributable to occupational exposures (20% in men and less than 3% in women) [5,6], an effort might be particularly required for men.

The SAQ response rate (37%) was lower compared to the pilot study (53%), not only overall, but also for the cancer center in Lyon (43% vs. 53%). The data suggest a self-censorship from current smoking patients, more easily attributing the origin of their disease to smoking, stressing the importance of providing information on occupational exposures to lung cancer patients. One of the purposes of the call reminder is to bring appropriate information to patients on occupational exposures related to lung cancer. Nevertheless, this may raise ethical questions, as it is crucial not to impose any coercion on the patient and that this process remains voluntary. In a previous study, we pointed out that this process could be perceived as an additional burden, and some patients need to focus their energy only on therapies. The expected benefit of the process may seem abstract and faraway in a context where patients experience difficulties to project themselves into the future [30].

Variations were also observed at every step of the process involving the physician (i.e., SAQ assessment, occupational cancer consultation). As there are no specific diagnosis features for occupation-related lung cancers, the medical judgment necessarily includes a subjective dimension and may differ according to the health professional. Indeed, some physicians made a more drastic SAQ selection, which led to a greater number of occupational disease declarations at the end of the consultation, whereas other professionals preferred to propose the occupational consultation to more patients to evaluate the possibility of an occupational disease during this consultation and thus delivered a smaller number of medical certificates.

Despite these differences, the monthly national multidisciplinary occupational cancer board helped to limit the discrepancies and to standardize the process to obtain data comparable to the literature in terms of work-related lung cancers [5,37].

In our study, patients experienced the same barriers as those previously identified in the literature and in the pilot study at each step of the process [24,25,27,29,30]. Yet, we added a 6-month follow-up performed by the social worker for patients who claimed for compensation, and we observed an improvement in the proportion of patients who have carried out the process (74% vs. 60% in the pilot study). We have reported on barriers to claim for compensation in previous studies [29,30]. The main barriers included the administrative burden and complexity of the process, patients’ fatigue and short life expectancy, as well as a perceived conflict of loyalty to the employer and/or reluctance to seek financial compensation. In the present study, information was not collected systematically on why patients did not claim for compensation. The social worker is a valuable resource to assist the patient in this long and complex administrative process and should probably be involved earlier in the screening process. Fatigue, in the context of oncological treatment, is one of the barriers often mentioned by patients who do not respond to the SAQ and who do not attend the occupational disease consultation [30]. Following the evolution of medical practices, in particular with the COVID-19 pandemic, teleconsulting appears as an opportunity to be able to carry out this type of consultation, limiting the risk of drop-out.

The clinical workload of oncologists and lung cancer specialists, as well as the time required for collecting the occupational history, make the screening of occupational exposures a difficult pursuit in the oncology encounter [14,23]. This barrier has been widely described in the literature [14,16,22,23], In addition to a lack of time, some physicians tend to pay little attention to occupational exposures as they do not affect the treatment process. Several interventions have been conducted to raise awareness among physicians on this issue, mainly through educational meetings [38,39,40]. Results were inconsistent and did not help to improve the reporting of occupational exposures, even with the introduction of mandatory requirements [39,40]. For these reasons, the present process relies on the identification of patients through the weekly multidisciplinary lung board and sending of the SAQ [27]. Of note, the process in place at the Centre Léon Bérard has led to enhanced collaboration between oncologists and occupational physicians, as well as increased awareness among oncologists, and to date numerous eligible patients have been directly addressed to the occupational cancer consultation prior to sending the SAQ.

In order to pursue the evaluation of the screening procedure in other medical situations, our process was also tested for other cancer sites (head and neck squamous cell carcinoma (HNSCC) and lymphoma) [41].

Despite its prohibition in France since 1997, asbestos was the predominant occupational carcinogen identified in study patients. Even though this carcinogen has been banned in most countries around the world, these results can be explained by asbestos’ long latency period [5,42] and an increased risk of lung cancer even with a low level of exposure [37]. A recent analysis in 702 patients enrolled in Lyon (i.e., participants from the pilot study and the present study) suggested a worse overall survival in lung cancer patients occupationally exposed to asbestos compared to patients non-exposed to asbestos [43]. If asbestos exposure negatively affects patient survival, the evaluation of occupational exposures becomes even more important in line with better understanding the impact of exposure on disease progression.

The widespread under-recognition of occupational causes of lung cancer is a significant barrier to lung cancer prevention. In the past few years, numerous lung cancer screening programs have been established in several industrialized countries [44]. Two large randomized controlled trials show that periodic low-dose chest CT scanning reduces lung cancer mortality [45,46], but to date, age and smoking are the main risk factors for lung cancer to be eligible for those programs. Nevertheless, studies were conducted to assess the efficiency of regular CT scan screening in asbestos-exposed workers. A French study demonstrated that this screening program is effective in detecting asymptomatic lung cancer in this population [47]. Another study has shown that screening of asbestos-exposed persons can be relevant but not in persons with no additional risk factors [48].

In 2017, a French working group made recommendations for the experimentation of low-dose scans for lung cancer screening in workers with a history of exposure to Group 1 IARC occupational lung carcinogens [49]. In France, the French National Authority for Health recommends a low-dose CT scan for workers exposed to asbestos for at least one year [50].

To date, the use of regular low-dose CT scans in the asbestos-exposed population has shown promising findings, but studies have mostly been modest in size, variable in design, and short-term in follow-up. It seems currently relevant to offer a regular low-dose CT screening for lung cancers to workers aged ≥50 years who have a history of five or more years of asbestos exposure, and it is now required to consider how to identify such workers and to organize screening programs with the help of public policy makers who have a key role in encouraging screening on a national level [44]. Our systematic screening process will help to reduce the under-reporting of occupational exposures in lung cancer and thus emphasize the need for screening programs in asbestos-exposed workers.

## 5. Conclusions

Our national study demonstrated that a systematic screening of occupational exposures is feasible and brings a substantial contribution to improve the detection of occupational exposures in lung cancer patients. As barriers still exist at every step of the process, it is essential to assist patients until the end of the process with the help of social workers. Although not well known to patients and health professionals, the reporting and compensation of occupational diseases are included in patients’ rights and contribute to the prevention of occupational risks. Given the estimation of the attributable fraction for lung cancer and considering that patients exposed to asbestos may have poorer survival, this systematic screening procedure for occupational exposures appears to be an effective opportunity to tackle this public health issue. Together with the cost assessment performed in a previous study [29], the results of the present study further provide important information of interest and of relevance to health policy actors to reduce occupational exposures, as well as to limit the complexity and administrative burden of the claim process in patients with work-related cancers.

## Figures and Tables

**Figure 1 ijerph-20-05068-f001:**
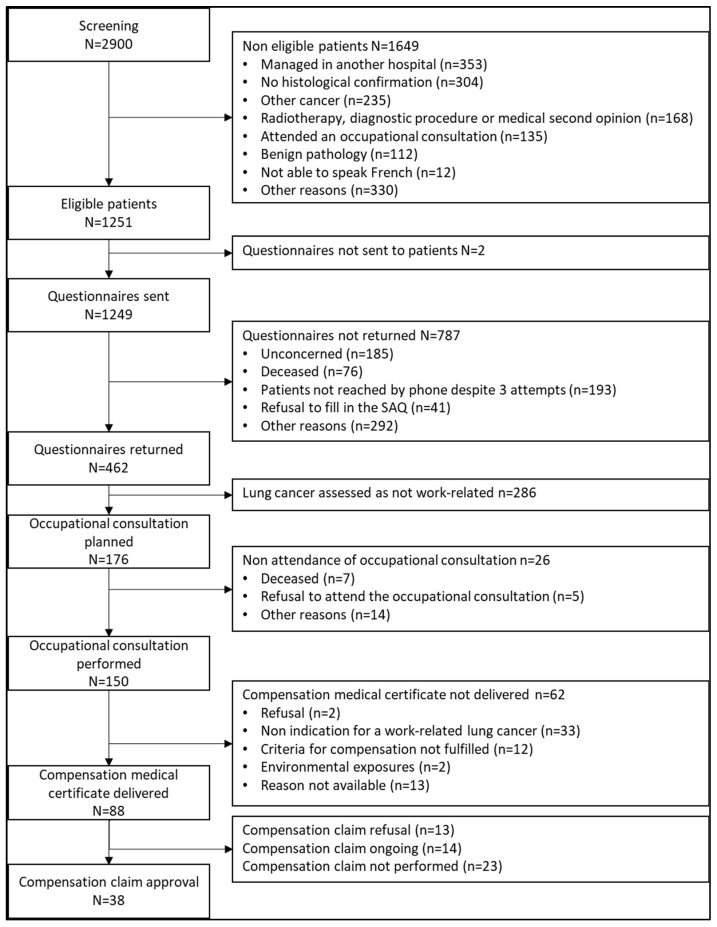
Study flowchart, 2017–2019, France.

**Table 1 ijerph-20-05068-t001:** Demographic characteristics of PROPOUMON national patients, globally and per center, 2017–2019, France.

	Overall (N = 1251)	Lyon (N = 444)	Caen (N = 175)	Lille (N = 167)	Clermont-Ferrand (N = 166)	Avignon (N = 299)	*p* Value
Sex							
Female	443 (35.4%)	166 (37.4%)	57 (32.6%)	62 (37.1%)	69 (41.6%)	89 (29.8%)	0.075
Male	808 (64.6%)	278 (62.6%)	118 (67.4%)	105 (62.9%)	97 (58.4%)	210 (70.2%)	
Age							
Mean (SD) [min-max]	66.48 ± 10.08 [29–94]	66.34 ± 10.56 [29–90]	67.83 ± 9.09 [41–94]	64.92 ± 9.80 [38–90]	66.64 ± 9.71 [33–88]	66.67 ± 10.22 [38–92]	0.1
Diploma							
No diploma	115 (26.0%)	47 (25.5%)	17 (27.4%)	18 (23.1%)	12 (22.6%)	21 (32.3%)	
General Certificate of Secondary Education	24 (5.4%)	4 (2.2%)	9 (14.6%)	4 (5.1%)	1 (1.9%)	6 (9.2%)	
BTEC diploma	134 (30.3%)	52 (28.3%)	23 (37.1%)	22 (28.2%)	19 (35.9%)	18 (27.7%)	
2-year university degree	40 (9.1%)	21 (11.4%)	3 (4.8%)	7 (9.0%)	4 (7.5%)	5 (7.7%)	
High school diploma or equivalent	72 (16.3%)	28 (15.2%)	7 (11.3%)	18 (23.1%)	11 (20.8%)	8 (12.3%)	
>2-year university degree	57 (12.9%)	32 (17.4%)	3 (4.8%)	9 (11.5%)	6 (11.3%)	7 (10.8%)	
Unknown	809 (64.6%)	260 (58.6%)	113 (64.6%)	89 (53.3%)	113 (68.1%)	234 (78.3%)	
Tobacco							<0.001
Non-smokers	178 (14.8%)	84 (19.0%)	20 (12.3%)	21 (12.6%)	30 (18.4%)	23 (8.7%)	
Former smokers	690 (57.5%)	251 (56.8%)	94 (58.0%)	92 (55.1%)	74 (45.4%)	179 (67.5%)	
Current smokers	331 (27.6%)	107 (24.2%)	48 (29.6%)	54 (32.3%)	59 (36.2%)	63 (23.8%)	
Unknown	52 (4.2%)	2 (0.5%)	13 (7.4%)	0 (0.0%)	3 (1.8%)	34 (11.4%)	
Number of pack-years							0.072
mean (SD) [min-max]	40.57 ± 20.85 [1–150]	38.30 ± 20.70 [1–120]	41.88 ± 20.23 [2–100]	40.55 ± 20.01 [4–150]	43.21 ± 20.97 [6–100]	45.32 ± 22.90 [11–104]	
Unknown	470 (37.6%)	93 (20.9%)	72 (41.1%)	37 (22.2%)	37 (22.3%)	231 (77.3%)	
EPICES ^1^ score							0.015
[0–30)	224 (63%)	102 (68%)	23 (46%)	33 (61%)	33 (77%)	33 (58%)	
[30–100]	129 (37%)	47 (32%)	27 (54%)	21 (39%)	10 (23%)	24 (42%)	
Unknown	898	295	125	113	123	242	

SD = standard deviation; min = minimum; max = maximum; BTEC = Business and Technology Education Council. ^1^ Evaluation of Deprivation and Inequalities in Health Examination Centers.

**Table 2 ijerph-20-05068-t002:** Clinical characteristics of PROPOUMON national patients, globally and per center, 2017–2019, France.

	Overall (N = 1251)	Lyon (N = 444)	Caen (N = 175)	Lille (N = 167)	Clermont-Ferrand (N = 166)	Avignon (N = 299)	*p* Value
Time of enrollment							<0.001
Newly diagnosed	712 (57.1%)	299 (67.6%)	111 (63.4%)	69 (41.3%)	54 (32.5%)	179 (60.1%)	
Follow-up	318 (25.5%)	39 (8.8%)	23 (13.1%)	60 (35.9%)	103 (62.0%)	93 (31.2%)	
Progressive disease	218 (17.5%)	104 (23.5%)	41 (23.4%)	38 (22.8%)	9 (5.4%)	26 (8.7%)	
Unknown	3 (0.2%)	2 (0.5%)	0 (0.0%)	0 (0.0%)	0 (0.0%)	1 (0.3%)	
Cancer stage ^1^							<0.001
1	142 (12.0%)	68 (16.5%)	7 (4.3%)	20 (13.2%)	35 (21.5%)	12 (4.0%)	
2	77 (6.5%)	33 (8.0%)	10 (6.1%)	6 (4.0%)	18 (11.0%)	10 (3.4%)	
3	302 (25.4%)	98 (23.8%)	48 (29.3%)	36 (23.8%)	38 (23.3%)	82 (27.5%)	
4	667 (56.1%)	213 (51.7%)	99 (60.4%)	89 (58.9%)	72 (44.2%)	194 (65.1%)	
Unknown	63 (5.0%)	32 (7.2%)	11 (6.3%)	16 (9.6%)	3 (1.8%)	1 (0.3%)	
Histology							
Adenocarcinoma	774 (62.2%)	282 (63.9%)	107 (61.5%)	104 (62.3%)	113 (68.1%)	168 (56.6%)	
Non-small cell lung carcinoma	93 (7.5%)	34 (7.7%)	6 (3.4%)	19 (11.4%)	13 (7.8%)	21 (7.1%)	
Small cell lung carcinoma	107 (8.6%)	33 (7.5%)	22 (12.6%)	13 (7.8%)	3 (1.8%)	36 (12.1%)	
Squamous cell carcinoma	246 (19.8%)	81 (18.4%)	39 (22.4%)	31 (18.6%)	30 (18.1%)	65 (21.9%)	
Other	25 (2%)	11 (2.5%)	0 (0.0%)	0 (0.0%)	7 (4.2%)	7 (2.4%)	
Unknown	6 (0.5%)	3 (0.7%)	1 (0.6%)	0 (0.0%)	0 (0.0%)	2 (0.7%)	
Previous cancer history	272 (21.8%)	121 (27.4%)	42 (24.0%)	61 (36.5%)	27 (16.3%)	21 (7.0%)	<0.001
Unknown	4 (0.3%)	3 (0.7%)	0 (0.0%)	0 (0.0%)	0 (0.0%)	1 (0.3%)	

^1^ TNM classification of malignant tumors 8th version; other = carcinoid, undifferentiated lung carcinoma.

**Table 3 ijerph-20-05068-t003:** Description of the systematic screening of occupational exposures, globally and per center, 2017–2019, France.

	Overall (N = 1251)	Lyon (N = 444)	Caen (N = 175)	Lille (N = 167)	Clermont-Ferrand (N = 166)	Avignon (N = 299)	*p* Value
SAQ ^1^ shipping	1249 (99.8%)	444 (100.0%)	175 (100.0%)	166 (99.4%)	166 (100.0%)	298 (99.7%)	
SAQ ^1^ return	462 (36.9%)	191 (43.0%)	64 (36.6%)	80 (47.9%)	58 (34.9%)	69 (23.1%)	<0.001
Occupational consultation proposed	176 (14.1%)	70 (15.8%)	24 (13.7%)	33 (19.8%)	11 (6.6%)	40 (55.1%)	0.001
Occupational consultation performed	150 (12.0%)	58 (13.1%)	17 (9.7%)	30 (18.0%)	9 (5.4%)	36 (13.4%)	0.3
Compensation medical certificate delivered	88 (7.0%)	24 (5.4%)	17 (9.7%)	15 (9.0%)	9 (5.4%)	23 (7.7%)	
Claim for compensation	65 (5.2%)	23 (5.2%)	9 (5.1%)	12 (7.2%)	8 (4.8%)	13 (4.3%)	
Compensation claim outcome							
Approval	38 (3.0%)	18 (4.0%)	4 (2.3%)	5 (3.0%)	2 (1.2%)	9 (3.0%)	
Refusal	13 (1.0%)	1 (0.2%)	4 (2.3%)	3 (1.8%)	1 (0.6%)	4 (1.3%)	
Ongoing	14 (1.1%)	4 (0.9%)	1 (0.6%)	4 (2.4%)	5 (3.0%)	0 (0.0%)	

^1^ SAQ = self-administered questionnaire.

**Table 4 ijerph-20-05068-t004:** Characteristics of the PROPOUMON national patients per SAQ ^1^ return status, 2017–2019, France.

	Overall (N = 1251)	SAQ Return		*p* Value
		Yes (N = 462)	No (N = 789)	
Age				0.001
<65 years	497 (40%)	340 (43%)	157 (34%)	
≥65 years	754 (60%)	449 (57%)	305 (66%)	
Sex				0.042
Female	443 (35%)	296 (38%)	147 (32%)	
Male	808 (65%)	493 (62%)	315 (68%)	
Time of enrollment				0.2
Newly diagnosed	712 (57%)	444 (56%)	268 (58%)	
Follow-up	318 (25%)	213 (27%)	105 (23%)	
Progressive disease	218 (17%)	131 (17%)	87 (19%)	
Unknown	3	1	2	
Cancer stage ^2^				0.028
I, II, III	521 (44%)	313 (41%)	208 (48%)	
IV	667 (56%)	442 (59%)	225 (52%)	
Unknown	63	34	29	
Histology				0.016
Adenocarcinoma	774 (62%)	504 (64%)	270 (59%)	
Non-small cell lung carcinoma	93 (7.5%)	55 (7%)	38 (8.3%)	
Small cell lung carcinoma	107 (8.6%)	73 (9.3)	34 (7.4%)	
Squamous cell carcinoma	246 (20%)	135 (17%)	111 (24%)	
Other	25 (2%)	19 (2.4%)	6 (1.3%)	
Unknown	6	3	3	
Tobacco				<0.001
Non-smokers	178 (14.8%)	94 (13%)	84 (19%)	
Former smokers	690 (57.5%)	416 (56%)	274 (61%)	
Current smokers	331 (27.6%)	238 (32%)	93 (21%)	
Unknown	52 (4.2%)	41	11	

^1^ SAQ = self-administered questionnaire; ^2^ TNM classification of malignant tumors 8th version; other = carcinoid, undifferentiated lung carcinoma.

**Table 5 ijerph-20-05068-t005:** Main occupational exposure retained (number and proportion) by the physician and claim for compensation outcome, 2017–2019, France.

Occupational Exposure Retained for Compensation	Compensation Medical Certificate Delivered (N = 88)		Claim for Compensation Approval (N = 38)	
Asbestos	79	89.8%	36	94.8%
Ionizing radiations	3	3.4%	1	2.6%
Coal tar, coal-tar pitch, coal combustion soot	2	2.2%	0	0.0%
Chromic acid, alkali or alkaline earth chromates and dichromates, zinc chromate	2	2.2%	0	0.0%
Iron ore	2	1.1%	0	0.0%
Diesel fumes	1	1.1%	1	2.6%

## Data Availability

The data presented in this study are available upon request from the corresponding author. The data are not publicly available due to privacy.

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
