# Peer review of "A Multicenter Study to Assess a Systematic Screening of Occupational Exposures in Lung Cancer Patients"

_ijerph, 2023, doi:10.3390/ijerph20065068_

Round 1
Reviewer 1 Report
The researchers did extensive work in a manuscript entitled " A multicenter study to assess a systematic screening of occupational exposures in lung cancer patients ". The authors conducted a survey of lung cancer detection and compensation. This is a subject of interest to researchers in related fields. I think this work is quite original in this field. Compared with other published materials, this study fills the current gap and can provide some help for improving occupational exposure detection in lung cancer patients. The background information is sufficient in the introduction. The article is coherent and logical. The conclusion is largely consistent with the evidence and arguments presented. The main problem is the lack of in-depth analysis of the survey results. For example, there is no further explanation in the article as to why there are cases where patients have not made a claim. What's behind it? How can this research be used to improve future medical and administrative procedures? If possible, it would be better to beautify the layout of some tables in the article. Try to have the words "Compensation medical certificate delivered" and "Claim for compensation approval" in Table 5 displayed in three lines instead of one word and one line as they are now.Author Response
Please see the attachment

Reviewer 2 Report
Dear Authors
First of all, I appreciate the extensive work carried out by the researchers on the study entitled "A multicenter study to assess a systematic screening of occupational exposures in lung cancer patients".
The background information, methodology are clearly explained and the discussion was well-written.
Your national study demonstrated that a systematic screening of occupational exposures is feasible and bring a significant contribution to improve the detection of occupational exposures in lung cancer patients.
I need minor clarifications:
- Regarding figure 1, there were 1251 eligible patients. Why the questionnaire was not sent to 2 patients? Give details.
- Compensation medical certificate was delivered to 88 patients. Why compensation claim was not performed in 23 patients? Patients who wanted to claim for compensation were offered help from a social worker for the administrative procedure. The social worker carried out a 6-month follow-up by phone to assess with the patient the state of progress of the compensation procedure and to provide further assistance if necessary. Still why the compensation claim was not performed. Explain.
Reviewer 3 Report
Dear authors
Thank you for allowing me to review your manuscript titled “A multicenter study to assess a systematic screening of occupational exposures in lung cancer patients”.
Congratulations for your effort to improve the under reported occupational diseases issue. From my point of view it is a relevant problem that deserves to claim out.
I will mention some considerations, questions and suggestions, in order to improve your manuscript.
Generally, please consider if all the self-references are essential.
In abstract, I suggest to include the study design. In the abstract conclusions it is mentioned “significant contribution” Why is is significant?
Materials and Methods,
In the study design authors describe it as a “prospective”, but I have the perception that the questionnaires are filled only once in a time. In this sense seems to me as a “transversal study”, with the data collected also from the occupational medical consultation. I understand that authors assess the “state of progress of the compensation procedure” along the time. Please clarify it.
Study population,
Please, specify if retired workers are included in the study. Was performed a post occupational suirveillance? In this sense, inclusion criteria should be specified.
About smoking, as you mention it as a non work related risk factors (line 160), was it a exclusion criteria? What about smokers workers exposed to asbesto?
Line 146, please define “individual deprivation”.
Statistical Analysis
I miss some multivariable analysis. For example, I wonder if the fact of no having academic degree (26%) has influence in the SAQ return and, consequently, in the claim for compensation.
Results
Please, in table 1 review the sum of percentages.
I suggest to eliminate table 4, because it is redundant with other tables and the text.
Discussion,
In lines 340-344 paragraph, I suggest to discuss from a gender perspective. Perhaps the effort might be particularly in women because an underdiagnosed lung cancers attributable to occupational exposures due to the lack of time to visit the medical services (“women doble presence”).
About the “few actions to improve this issue” (line 331) I recomend to read the following reference article: Benavides FG, Ramada JM, Ubalde-Lopez M, Delclos GL, Serra C. A hospital occupational diseases unit: an experience to improve the identification and recognition of occupational disease. Med Lav, 2019;110,4: 278-284. DOI: 10.23749/mdl.v110i4.8138
Line 389, please define CLB 389
Limitations should be mentioned. For example, the sample looks like a convenience one, not aleatory. Or the fact of having excluded not frech speakers workers…
Conclusions,
“... the process cost is largely acceptable...” Did authors measure the costs in the study?
Please, limit it to your findings acording to the study aims.
